# Amphibious epidermal area networks for uninterrupted wireless data and power transfer

Amirhossein Hajiaghajani [1,4], Patrick Rwei[2,4], Amir Hosein Afandizadeh Zargari[1,3], Alberto Ranier Escobar [2], Fadi Kurdahi[1,3], Michelle Khine[2] & Peter Tseng [1,2] ✉

The human body exhibits complex, spatially distributed chemo-electro-mechanical processes that must be properly captured for emerging applications in virtual/augmented reality, precision health, activity monitoring, bionics, and more. A key factor in enabling such applications involves the seamless integration of multipurpose wearable sensors across the human body in different environments, spanning from indoor settings to outdoor landscapes. Here, we report a versatile epidermal body area network ecosystem that enables wireless power and data transmission to and from battery-free wearable sensors with continuous functionality from dry to underwater settings. This is achieved through an artificial near field propagation across the chain of biocompatible, magneto-inductive metamaterials in the form of stretchable waterborne skin patches—these are fully compatible with pre-existing consumer electronics. Our approach offers uninterrupted, self-powered communication for human status monitoring in harsh environments where traditional wireless solutions (such as Bluetooth, Wi-Fi or cellular) are unable to communicate reliably.

Body area networks (BANs) enable the interconnection of wearable microelectronic nodes across various human body areas and are utilized in consumer electronics, the Internet of things, and advanced healthcare[1-4]. These wearable nodes may consist of sensors, peripherals, smart accessories, or custom biomedical devices[5]. The functionality of these networks heavily depends on user comfort, compatibility with pre-existing cyber-physical local area networks, and seamless body integration under a diverse range of environmental settings entailing homes, offices, or vehicles in addition to various climates such as dry, humid, or underwater environments[6-9].

Traditional sensor area networks have been implemented by daisy-chaining a central controller hub and peripheral sensing nodes through wires to transfer power and data[10-12]. The robustness of these systems, however, significantly declines by utilizing multiple connectors, particularly in moist or underwater environments, which leads to end-user discomfort in wearable applications[13]. Difficult network expansion, lack of mobility, and reliance on physical connectors have restricted the wired platforms to controlled clinical settings[14]. On the other hand, end-users benefit from wireless technologies such as Bluetooth, Wi-Fi, or cellular that enable high-throughput communication[15-17]. The far-field nature of electromagnetic wave propagation, however, renders these technologies inefficient in extremely humid or underwater settings due to notably low penetration depth at ultra-high frequency radio bands[17-19]. Additionally, near-field magnetically coupled antennas have demonstrated significantly improved path loss in the proximity of the human body compared to capacitive body-coupled and far-field radiative approaches, however, such technologies are not yet compatible with pre-existing communication

[1]Department of Electrical Engineering and Computer Science, University of California, Irvine, CA, USA. [2]Department of Biomedical Engineering, University of California, Irvine, CA, USA. [3]Center for Embedded and Cyber-physical Systems, University of California, Irvine, CA, USA. [4]These authors contributed equally: Amirhossein Hajiaghajani, Patrick Rwei. ✉e-mail: tsengpc@uci.edu

protocols, therefore not readily available for wearable electronics[20,21]. The availability of hardware modules and software interfaces can critically limit BAN usage in emerging wearable applications that aim to minimize end-user costs while integrating the multitude of third-party sensors and devices. Alternative wireless solutions include extremely low or high frequency bands which possess high latency, or incompatibility with consumer electronics[22–25].

Importantly, these wireless technologies do not support power harvesting for peripheral nodes, leaving wearable electronics reliant on batteries. Despite recent advances in wearable batteries[26,27] and power scavenging approaches[15,28], the chemicals utilized in these batteries are often bio-incompatible or exhibit limited lifespan with relatively large or fragile form factors. Moreover, increased environment humidity degrades body-coupled capacitive power transfer[29–32]. These factors often impose critical constraints on sensitive biomedical wearables, especially in terms of requiring continuous power and data transfer in dynamically unpredictable environments. Among widely available wireless technologies, only radio frequency identification and notably near field communication (NFC) feature dedicated power transfer to battery-free peripheral electronics in the immediate proximity of the untethered controller[33,34]. Despite these unique characteristics, utilization of electromagnetic near fields is typically impeded in BANs due to inadequate short range (about a few centimeters) to reach many different human body areas.

There have been recent advances in pushing the NFC range to relatively longer distances (about a few feet) for wearable and/or epidermal solutions[7], such as embroidering inductor-hubs into clothing[35] or skin patches[36] to direct electromagnetic emission on different body areas. However such inductor hubs are difficult to integrate into clothing, restrict the wearable's placement to a dedicated hub location, and impose excessive path loss[35]. To mitigate these challenges, our team introduced NFC-compliant, textile-integrated magnetic metamaterials with hub-free BAN architecture that connects wearable electronics through discrete pieces of clothing[37]. These solutions, however, utilize conductive threads that exhibit relatively large conduction loss[35], copper sheets that lack in stretchability[37], polymer-encapsulated liquid metal substances that exhibit questionable biocompatibility[36], or metallic nanofibers with difficult synthesis and microelectronic integration (utilizing cleanroom facilities)[34,38,39]—these are all challenging for on-skin, epidermal settings. In addition, these solutions generally function in controlled environments and can be easily disturbed in naturally diverse outdoor landscapes. A versatile multi-environment wireless network would be able to offer seamless human interaction with the wearable electronics that are increasingly utilized across health, body-monitoring, and emerging virtual/augmented reality-based technologies.

Here, we report waterborne, biocompatible epidermal skin patches to enable wireless power and data transfer on the human skin and across various body areas within a very diverse environmental setting spanning from dry indoors to underwater outdoors. This BAN is based on a chain of magnetically coupled resonators that exhibits artificial nearfield wave propagation across the resonator array known as the magneto-inductive (MI) metamaterials. Our maximally bio-friendly strategy introduces a new metamaterial geometry and paint-on-skin silver flake-based ink to achieve a balance between electrical conductivity, mechanical stretchability, and flexibility. The higher conductivity and mechanical stability with nontoxic polymer binders are achieved by shifting from solvent-based[40–42] to waterborne paint, in addition to utilizing silver over carbon flakes[43,44]. These epidermal patches are readily transferred onto the skin and linked through NFC-compatible battery-free electronics by enabling nearfield electromagnetic wave propagation across the metamaterial to different body areas. The easy fabrication approach and biocompatible nature of the introduced ink establish an on-demand yet seamless communication between untethered battery-free sensors and an active controller (such as water-resistant smartwatches with growing battery capacities) and span through nearby people and objects to facilitate human interaction with cloud-assisted electronics.

## Results and discussion
### Multi-environment network design
MI metamaterials are suitable candidates for near-field electromagnetic (EM) waveguide applications, as they allow propagation across a chain of magnetically coupled resonant structures within a particular passband[45,46]. Additionally, these metamaterials demonstrate highly engineered mechanical and spectral EM properties, allowing for efficient power transfer and data reception[37]. Due to the insensitivity of the magnetic interconnections (between the resonators) to the electrical permittivity of the surrounding environment, the MI metamaterials exhibit a notable degree of spectral immunity to changes in the background media (Fig. 1a).

Here, we demonstrate epidermal MI metamaterial networks to realize a BAN that is directly placed on the skin and functions seamlessly and regardless of the user's choice of clothing. Our network is formed by arrays of magnetically coupled resonators that consist of a multiturn planar loop with a self-inductance and ohmic resistance of $L$ and $R$, respectively. The resonance properties of this building block are determined by the effective capacitance which consists of the structural stray and a discrete tuning component ($C$). The stray capacitance forms in between the multiturn loops through the background environment ($C_B$) and skin ($C_S$), and relates to the coil's real estate. The lossy dielectric characteristic in the background and skin media is represented by $G_B$ and $G_S$, respectively. This lossy behavior, however, can be controlled by encapsulating the loop traces within a thin nonconductive insulator to block the fringing electric currents and is modeled by an equivalent insulating resistance ($R_{INS}$) series with the stray components (Fig. 1b).

Due to the parasitic nature of the stray electric fields, we incorporate a lumped surface mount capacitor to dominate the stray capacitance (larger in an order of magnitude) to achieve a steady resonance insensitive to skin proximity. This allows tuning the multiturn loop resonance into the NFC band's frequency of 13.56 MHz. This MI array can be excited through a magnetically coupled external source, to enable flowing sinusoidal time-dependent electric currents with an angular frequency of $\omega$ in the $n$th loop resonator (ranging from 1 to $N$). The encapsulated magnetically coupled resonator coils are inductively coupled to the closest neighbor resonator with the mutual coupling of $M = k \times L$ where $M$ and $k$ represent the mutual inductance and coupling factor, respectively. Here, neighbor coils are positioned with small overlap (where coils are distanced by $d_c$) to realize a sufficient magnetic coupling. Traditionally, the dispersion characteristics of MI metamaterials have been calculated through the resonator impedance, followed by calculating the $n$th inductor current in a linear array represented by:

$$\mathbf{I_n} = I_1 e^{j\phi_1} e^{-j\gamma(n-1)d_c} \tag{1}$$

where $\phi_1$ and $I_1$ are determined by the boundary and excitation conditions, and $\gamma$ represents the propagation constant. Although this approach has successfully described the dispersion properties of simple resonators (such as planar multiturn loops), it may become excessively arduous to model complex resonating structures. Here, we utilize a systematic approach to derive the dispersion profiles of complex resonator structures via breaking the complex equivalent circuit of each resonator (unit cell) into sub-models whose transfer (known as ABCD) matrix models are simply cascaded:

$$\mathbf{T_{cell}} = \begin{bmatrix} A_{cell} & B_{cell} \\ C_{cell} & D_{cell} \end{bmatrix} = \mathbf{T_M} \times \mathbf{T_R} \times \mathbf{T_C} \tag{2}$$

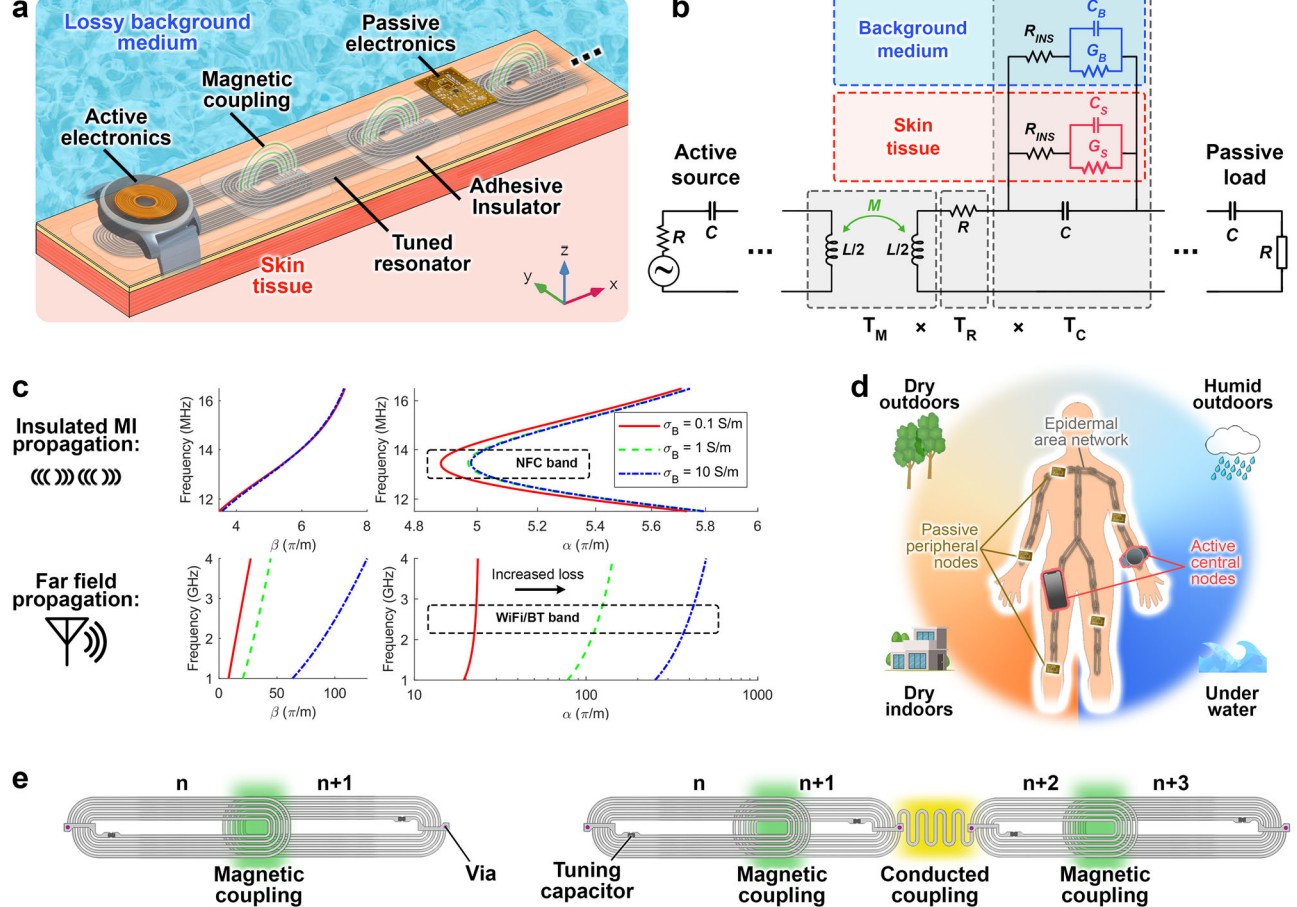

**Fig. 1 | MI waveguides for multi-environment wireless body area networks.**
**a** Schematic of MI wave propagation across magnetically-coupled epidermal skin patches exposed to lossy unpredicted environments. **b** Equivalent circuit of the resonator chain incorporating the background medium and insulator characteristics effect. **c** Dispersion diagrams of nearfield MI against far-field propagations. In contrast to conventional far-field propagation, MI propagation exhibits little to no sensitivity to lossy background media. **d** This makes MI-BAN a suitable candidate for multi-environment area networks, spanning consumer wireless communication and powering from lossless dry to lossy underwater environments. **e** Rectangular resonators can be connected through wireless magnetic or wired conducted coupling (with serpentine structure) to enable maximal mechanical stretchability at frequently bending joints. Source data are provided as a Source Data file.

The resonator's sub-models transfer matrices here include $\mathbf{T_M}$, $\mathbf{T_R}$, and $\mathbf{T_C}$ that respectively represent the mutual inductive coupling, multi-turn loop Ohmic (DC) resistance, and tuning capacitor within the lossy environment (see "Methods" for transfer matrix calculation). The dispersion characteristics of the array of insulated coils are therefore extracted by deriving the eigenvalues of the unit cell's transfer matrix[47] with regard to the geometrical distance between neighbor cells:

$$\gamma = \frac{1}{d_c} \times \cosh^{-1}\left(\frac{A_{cell} + D_{cell}}{2}\right) \qquad (3)$$

where $\gamma = \beta + j\alpha$ ($\beta$ as the phase and $\alpha$ as attenuation constants). The propagation constant of the proposed encapsulated MI metamaterial in free space and lossy background media (at the NFC band) is analytically modeled and compared with that of the conventional far-field propagation (at Bluetooth and Wi-Fi band)[48]. Additionally, the overall transfer matrix of a linear MI array of N resonators ($\mathbf{T_{cell}^N}$) can be simply converted to the transmission profile in the form of end-to-end S-parameters[49]. Interestingly, the magnetic nature of the coupling in the MI metamaterials offers significantly lower sensitivity to the lossy surrounding environment represented by the electric conductivity of $\sigma_B$ (Fig. 1c). The bandwidth and attenuation at the passband which is directly correlated with the metamaterial transmission profile can be engineered by adjusting the coupling factor and equivalent encapsulation insulation (Supplementary Fig. 1).

A nearby electrically conductive object would bypass the fringe electric fields by offering a low-loss shortcut path, which can be modeled by shorting the fringe capacitors in the background medium ($C_B$). This may not only potentially shift the resonance characteristics of the coil, but also may diminish the transmission profile of the MI array. It justifies the need for encapsulating the resonators with appropriate insulation properties which is modeled as $R_{INS}$. However, due to mechanical limitations on insulator thickness, the insulation may not be ideal, hence a finite $R_{INS}$ value would be inevitable. This may induce slight resonance sensitivity to external metallic objects (which is controlled by a dominant lumped capacitor). Larger $R_{INS}$ (or thicker encapsulation) and lumped $C$ values warrant better immunity to such interferences by allowing fringing electric fields to form within the lossless insulator's thickness.

The low-loss propagation characteristics of this structure demonstrated at the tuned pass band allow untethered links within lossy environments, ultimately enabling an amphibious wireless power and signal transmission (Fig. 1d). These discrete resonators, however, have been traditionally linked by magnetic coupling. To demonstrate maximal mechanical stretchability across the MI metamaterial pathway, we introduce conducted-coupling of neighboring coils through a serpentine structure with a trace geometry that is optimized for frequently bending joints (Fig. 1e). The dispersion properties of these resonators are systematically analyzed by dividing the structure into sub-geometries and compared in Supplementary Fig. 2. The coil

geometries and tuning capacitors designed in this study are demonstrated in Supplementary Fig. 3. The overall performance of the proposed method is compared with other BAN solutions in Supplementary Table 1.

**Stretchable resonator coil fabrication**

The lack of stretchability in flexible copper-based printed circuit boards renders them unsuitable for biomedical applications, particularly when applied in epidermal settings. In contrast, silver conforms to various types of stretching surfaces including living tissues and/or a wide range of textiles[50]. Although the conductivity of the silver-based ink can be readily enhanced by increasing the concentration of silver flake, this results in lower flexibility and weaker stretchability properties[51,52]. Additionally, these inks are conventionally hindered by significant loss of conductivity under normal strain[53].

Traditional ink-jet printed silver inks require high-temperature sintering to be conductive. Here we introduce an alternative waterborne-based silver ink formulation with low curing temperature to balance the tradeoff between conductivity and stretchability. Our stretchable conductive ink consists of commercially available multi-purpose liquid glue as a matrix, silver flakes as a conductive agent, glyceryl triacetate as a plasticizer, and borax as crosslinkers sintered in low temperatures. In contrast to solvent-based conductive inks, the waterborne nature of the ingredients offers biocompatibility, faster drying time, and adhesion to a wide range of substances including skin and polymer substrates (see "Methods" for silver ink synthesis). Interestingly, these silver ink-based circuit traces bond with metallic pins of electronic components and eliminate the need for brittle solder connections that often fail in highly flexible settings of wearable applications. Additionally, the solder-free curing process enables the use of circuit board substrates with excellent mechanical flexibility that often possess sensitive thermal characteristics[54,55].

The planar coil designs (see Supplementary Fig. 3) were laser cut on a vinyl sheet which was used as a stencil placed on a thin PDMS sheet. Next, the silver ink was applied over the stencil, which was then removed to develop the coil pattern. Following this step, the surface mount tuning capacitor was placed on the dedicated area, and the coil batch was pre-cured in the oven. After the initial curing, the ends of the multiturn loop were connected by bending one of the coil ends over an insulating interlayer. The overlapping ends were then connected by an additional drop of silver ink followed by post curing (Fig. 2a). For coil fabrication details see "Methods".

In order to prevent direct contact with the background medium, a thin insulating PDMS sheet was used to encapsulate the coil. The transmission profile ($S_{21}$) of a 90 cm long MI array was then measured under various PDMS encapsulation thicknesses and background environments including lossless dry and 35 g/L NaCl to mimic ocean water salinity. The insulated array was submerged in the lossy background and transmission profiles were measured using a vector network analyzer (Fig. 2b). Here, a PDMS thickness of 0.5 mm demonstrates a path loss enhancement of 4.4 dB/m compared to 0.25 mm thick insulator. This improvement diminishes to 1.1 dB/m between PDMS thicknesses from 0.5 to 0.75 mm. To reach an optimal balance between the resonator's thickness (influencing flexibility and user comfort) and the transmission performance of the channel, we opted for PDMS sheets with a thickness of 0.5 mm for encapsulation.

The electrical conductivity of the silver flake-based ink was measured under an increasing strain (until failure) exhibiting resistance stability under applied strain which makes it suitable for use in epidermal applications (Supplementary Fig. 4). Additionally, the resonance characteristics (including the frequency and quality factor) of the designed rectangular coil was measured under normal strain (Fig. 2c) which demonstrates a hysteresis-free profile under up to 15% strain to represent a normal tissue bending (Fig. 2d). The extreme stretchability test of the encapsulated resonator demonstrates up to

120% of applied strain without mechanical failure (Fig. 2e). The not fully-encapsulated coil structure (with PDMS substrate and exposed ink) failed at a breaking point near the center of its length while under 25% strain. The resonant structure, however, did not fail at the conjunction of the tuning capacitor and the coil trace for up to 100% applied strain (see Supplementary Fig. 5).

For reliability purposes, the flexibility of the skin patch was tested by measuring the resonance characteristics ($S_{11}$) under up to 10,000 full bending (0–180°) cycles with no major changes in the central frequency (Fig. 2f–h). The overall performance of the proposed stretchable ink is compared with other conductive ink solutions in Supplementary Table 2.

The epidermal BAN is formed by placing the encapsulated resonant coils directly on the skin. Adjusting the overlap between the neighboring coils (via the mutual inductance $M$) enables engineering the passband's bandwidth. To this aim, a coil overlapping (about 2 cm on a 13 cm long coil) would induce a bandwidth greater than 848 kHz to comply with the highest data rate incorporated in the amplitude-shift keying modulation embedded in the NFC protocol. This translates to a 15% horizontal overlap between two neighbor resonators which is found through benchtop experiments. Additionally, a larger bandwidth ensures that a minor shift of resonance under bending (characterized in Fig. 2g) will be covered within the exhibited passband. The vertical distance between neighbor resonators here consists of the overall thickness of PDMS sub- and superstrate and is studied in Supplementary Fig. 6.

Here, we start forming the BAN by placing the serpentine structure on the joints (such as shoulders, elbows, and knees) and continue placing the rectangular coils to create the pathway. Due to the Ohmic resistance in conductively coupled resonators, the rectangular coils with magnetic coupling demonstrate slightly lower attenuation (as shown in Supplementary Fig. 2) and thus are prioritized across the non-bending areas of the BAN. The coils were secured in place using a self-adhesive transparent film (Fig. 3a). Here, the end-to-end transmission profile of a 45 cm long epidermal MI array (placed on the arm) is measured using two multiturn loop antennas (connected to a two-port vector network analyzer) under variable bending and twisting arm gestures (Fig. 3b). The passband stability during these movements warrants reliable wireless communication throughout the activity monitoring.

The specific absorption rate (SAR) for the epidermal MI metamaterial array is simulated under lossless and lossy background media (see Supplementary Fig. 7). According to the NFC standard specification, the peak magnetic field emitted from the active NFC reader may not exceed 10.5 A/m. This translates to a maximum simulated SAR of 81 mW/kg averaged over the muscle volume, which is significantly lower than the industry standard of 400 mW/kg[56–60] at 13.5 MHz.

Here we demonstrate the mid-range uninterrupted wireless power and data-sharing mechanism through the epidermal BAN in a wide span of environments including underwater settings. We placed an array of rectangular and serpentine coils on the skin, and ultimately secured them in place using transparent adhesive film (see "Methods" for implementation details). We utilized our custom-designed passive sensors with integrated external strain gauges and an on-board miniature loop antenna (that aligns with the BAN dimensions). Due to battery-free operation, these sensors are notably lightweight (3 g per board), highly flexible, and conform to the body areas, which realizes seamless human activity monitoring without interrupting the user's normal routine. The sensors were also encapsulated for underwater usage (see "Methods" for encapsulation details). Additionally, the NFC reader which incorporates a miniature battery represents the only active microelectronics that supplies power to the sensors and receives sensor-collected data for local or online storage. The miniature form factor of this controller enables its utilization as an NFC-enabled master node for future wearable applications such as

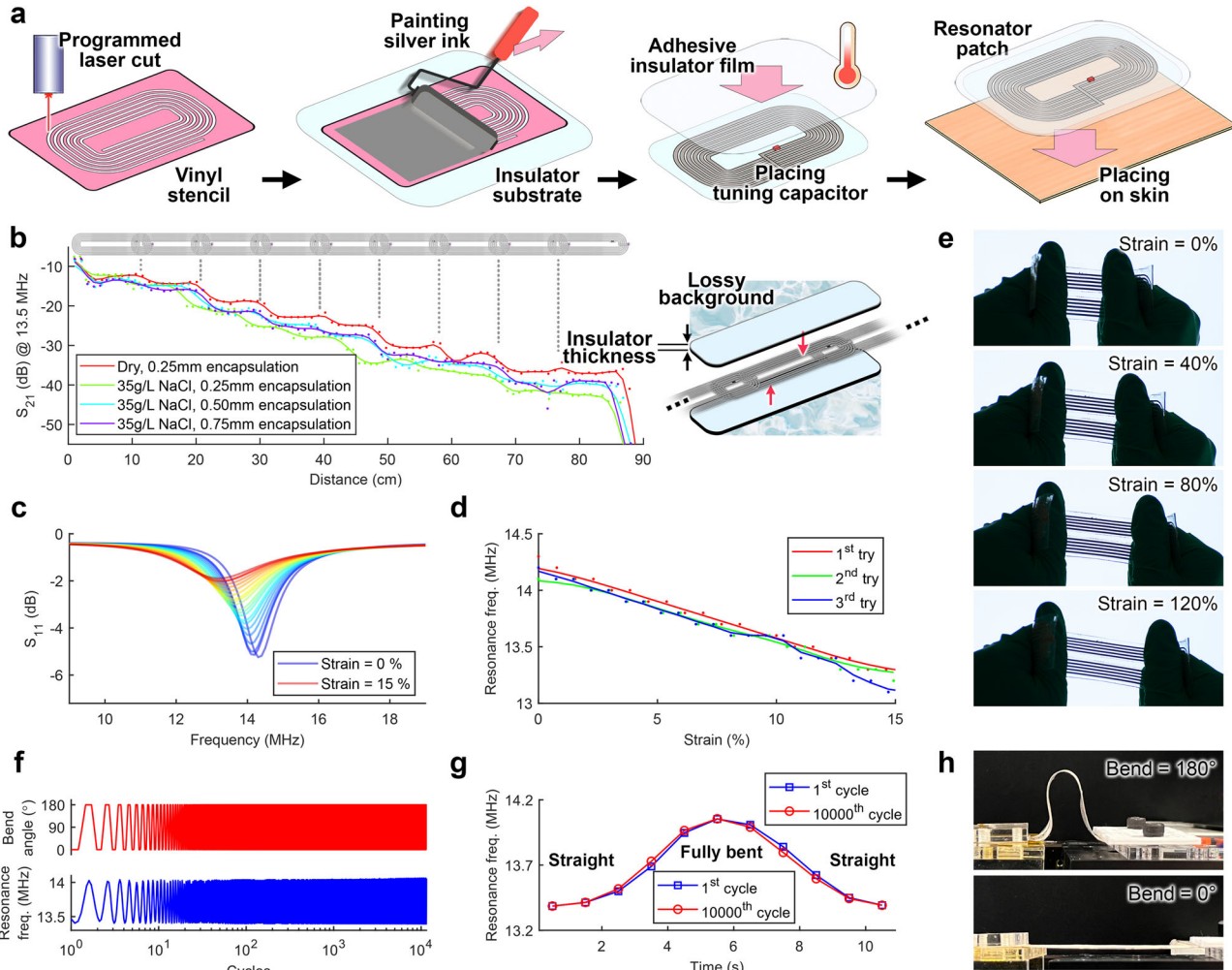

**Fig. 2 | Mechanical and electrical stability of the resonators. a** Fabrication of stretchable silver ink-based resonators and translation to skin. **b** Transmission profile of the MI array measured under various encapsulation thicknesses (insulating top and bottom of the skin patch) and background medium loss. **c** Resonance properties of the silver ink-based coil under normal strain in a dry background environment on a 0.5 mm thick PDMS substrate. **d** Sensitivity and repeatability of the resonant coil under normal strain. **e** Demonstration of mechanical stretchability of a fully encapsulated resonator under excessive applied strain. **f** Reliability test of flexibility under up to 10,000 full bending cycles. **g**, **h** The coil exhibits no major difference between the resonance frequency profile of the first and last bending cycles. Source data are provided as a Source Data file.

waterproof smartwatches (Fig. 4a, b). The printed circuit board layout of the custom reader and sensor boards are shown in Supplementary Fig. 8. In addition to on-demand integration of this BAN with pre-existing NFC-enabled electronics, the nearfield propagation can be seamlessly routed along and between the epidermal and textile-integrated form factors, enabling multi-purpose applications (Supplementary Fig. 9).

The reader was set to collect sensor data with an overall refreshing rate of 14 Hz (equivalent to 4.66 Hz per sensor). We employed software-based time domain multiple access for seamless switching between sensing nodes. This method utilizes controlled surface propagation of MI waves, which in contrast to wired or antenna switching schemes does not require additional multiplexing hardware. As a result, conventional NFC-enabled smartphones can operate as compatible readers. The raw digitized strain gauge values were translated to angular postures through a one-time calibration process by linearly correlating the measured sensor value to the known body angle.

The arm activity was recorded during periodic sequences of light swimming in an outdoor pool. The collected sensor data was smoothed through a real-time filter and compared against video reference. To assess the wireless efficiency of the network, we measured the NFC packet reception ratio (PRR) which represents the number of successfully received packets by the reader compared to the total number of requested packets. Here, a packet contains sensor information from all transponders within the network during a single refresh (Fig. 4c). Generally, conventional computer vision packages sporadically fail to extract the swimming posture angles particularly due to lack of mobility (causing misaligned camera position compared to the dynamically moving body), in addition to environmental arti-facts such as underwater reflections that interfere with the camera sensor (Supplementary Fig. 10).

Furthermore, the smooth transition and reliable performance of this NFC-based BAN are compared with standard Bluetooth-based solutions in underwater settings. While gradual wetting and full sub-mersion of this epidermal BAN causes 3% packet loss (at 10 Hz of total refreshing rate), the received signal strength indicator (RSSI) of the Bluetooth-based communication fails instantaneously by submerging the Bluetooth beacon only (see "Methods" for Bluetooth experiment details). This highlights the network's robustness and adaptability to different environments with varying levels of humidity and back-ground loss (Fig. 4d, e).

Furthermore, the long-term evaluation of the proposed power and communication link in underwater conditions showcases con-sistent PRR exceeding 89% (see Supplementary Fig. 11). This

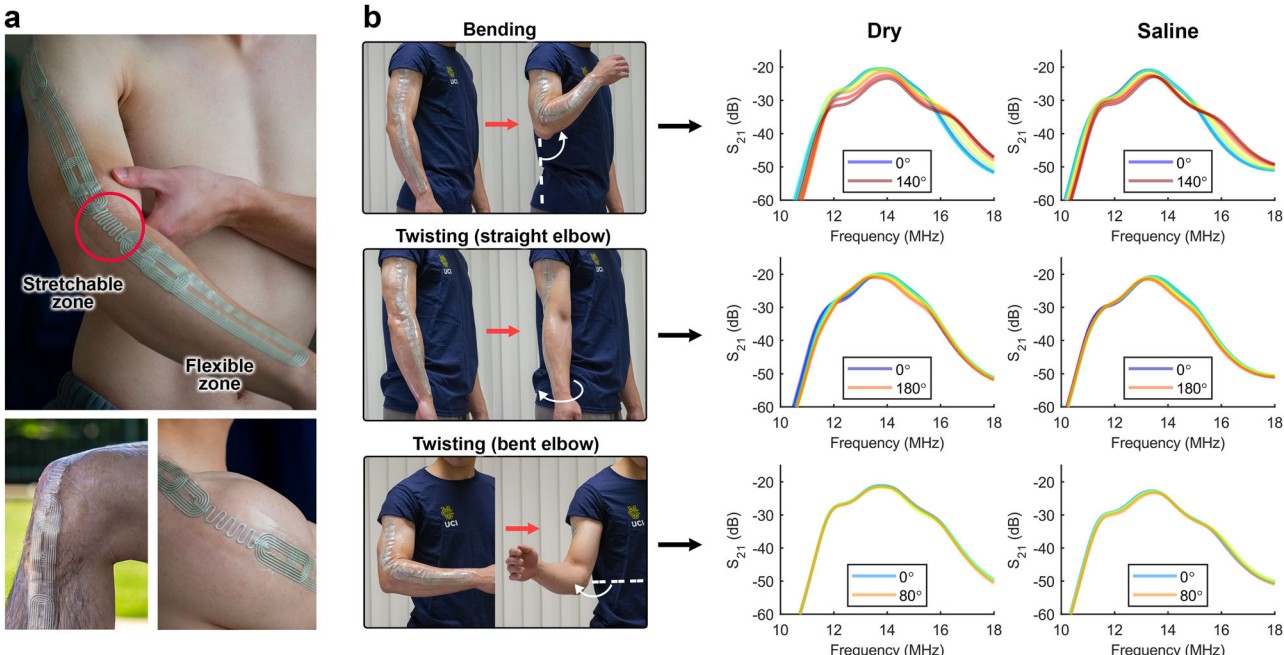

**Fig. 3 | Impact of body gestures on the stability of the skin-integrated network. a** MI metamaterial network transferred onto the skin with a stretch-friendly serpentine structure placed on the joints. **b** Transmission characteristics of the epidermal MI BAN under various joint movements exhibit a stable electromagnetic performance under dry and saline-coated (35 g/L NaCl) background media. Source data are provided as a Source Data file.

exceptional stability establishes a notably reliable link, setting it apart from conventional wireless protocols. Additionally, when considering environmental factors such as user sweating or the presence of metallic objects in proximity to skin patch resonators during outdoor use, the PRR remains consistently above 92% (Supplementary Fig. 12).

Importantly, standalone power harvesting mechanisms that do not rely on batteries, may benefit from this NFC-compatible BAN strategy that can efficiently distribute harvested power to distributed untethered loads, especially in challenging-to-reach places where lightweight solutions are crucial. Furthermore, this approach enables encapsulated MI skin patches to be utilized in a variety of hardware architectures, ultimately enabling wearable system designers to incorporate connector-free daughter boards (such as the wearable passive sensor boards here) within a larger platform. The hardware architecture of the encapsulated MI channels is compared with common wired approaches in Supplementary Fig. 13, showcasing the promising application of these units in multi-environment settings.

We have reported an epidermal nearfield BAN comprised of skin patches, enabling wireless power and data transmission to NFC-enabled electronics placed on various body areas in a significantly wide span of environmental settings, including underwater. This is the first report of a mid-range underwater wireless power and data transmission platform compatible with pre-existing consumer electronics for uninterrupted human activity and health monitoring. With recent technological progress that leaves little room to improve sensors' core functionality, wearable devices aim to address critical challenges such as minimizing power consumption, optimizing user comfort, and establishing seamless data flow between organic media and computer cloud systems. This study contributes significantly to this ongoing paradigm shift. We expect that the self-powering and multi-environment nature of this network could allow the concept to be extended in a number of directions. The notably lightweight battery-free sensors can be integrated with emerging augmented/virtual reality devices and could allow electronic components to be built alongside the body. The network's resilience to background media characteristics offers the potential for activity monitoring and an immersive user experience even in challenging and unpredictable dynamic settings, where Bluetooth, Wi-Fi, or cellular technologies fail.

## Methods

### Ethics statement
This research complies with all relevant ethical regulations, reviewed and approved in accordance with the University of California Irvine Institutional Review Board protocol HS#2018-4843. Consent has been obtained from the human subject (one 31-year-old male) involved in all demonstration experiments. The subject consents to appear in all figures presented in this study.

### Transfer matrix calculation of resonators with magnetic coupling
The entire resonator's ABCD matrix is derived by cascading the sub-model matrices shown in Fig. 1b. These matrices are each modeled via MATLAB's Symbolic Math Toolbox and are derived as follows:

$$\mathbf{T_{cell}} = \begin{bmatrix} A_{cell} & B_{cell} \\ C_{cell} & D_{cell} \end{bmatrix} = \mathbf{T_M} \times \mathbf{T_R} \times \mathbf{T_C}$$

$$\mathbf{T_M} = \begin{bmatrix} \frac{L}{2M} & j\omega\left(\frac{L^2}{4M} - M\right) \\ \frac{1}{j\omega M} & \frac{L}{2M} \end{bmatrix}, \mathbf{T_R} = \begin{bmatrix} 1 & R \\ 0 & 1 \end{bmatrix}, \mathbf{T_C} = \begin{bmatrix} 1 & \left(Z_S^{-1} + Z_B^{-1} + j\omega C\right)^{-1} \\ 0 & 1 \end{bmatrix},$$

where $Z_S = R_{INS} + \left(G_S + j\omega C_s\right)^{-1}$ and $Z_B = R_{INS} + \left(G_B + j\omega C_B\right)^{-1}$.

### Transfer matrix calculation of resonators with conducted coupling
Similar to magnetically coupled resonators, the entire unit cell's transfer matrix here is calculated using the equivalent circuit shown in Supplementary Fig. 2. Here, $\mathbf{T_M}$ is slightly different than the previous case as due to the nature of conducted coupling, each multiturn loop is entirely coupled to its neighbor loop on one side only. Here, $R_W$

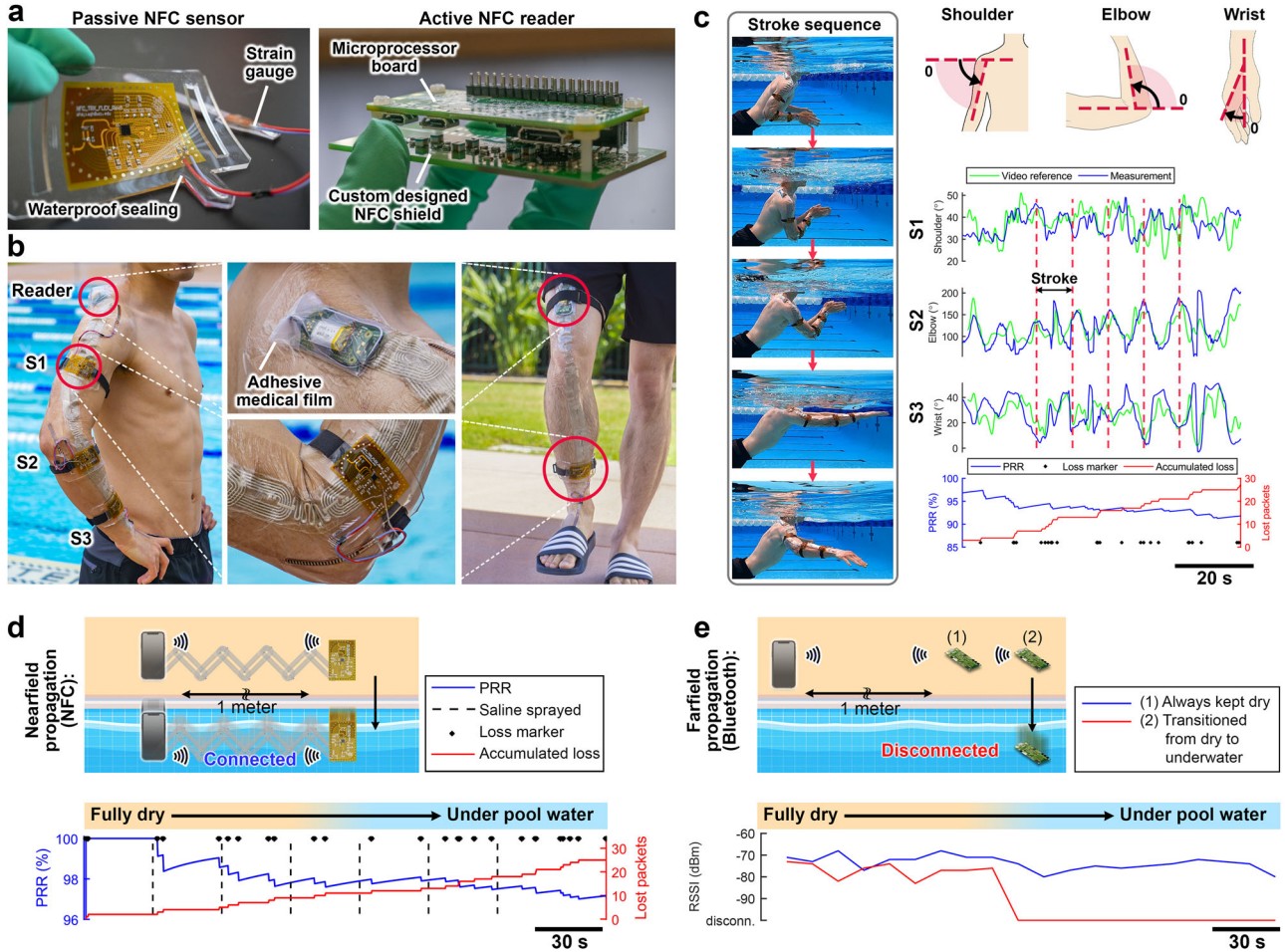

**Fig. 4 | Amphibius epidermal BAN functioning in a wide span of challenging environments. a** The passive flexible sensor board with an integrated strain gauge and miniature active NFC reader. **b** Integration of epidermal skin patches on the human body to transmit wireless power from the NFC reader to the local sensors and collect motion data. **c** Collected motion data from the strain gauges while swimming underwater. **d** The nearfield propagation of the NFC-based BAN continues to function reliably from dry to underwater environments. **e** Bluetooth communication fails immediately after submerging any of the nodes underwater. Source data are provided as a Source Data file.

represents the Ohmic loss of the serpentine wired coupling.

$$\mathbf{T_{cell}} = \begin{bmatrix} A_{cell} & B_{cell} \\ C_{cell} & D_{cell} \end{bmatrix} = \mathbf{T_M} \times \mathbf{T_R} \times \mathbf{T_C} \times \mathbf{T_W} \times \mathbf{T_C} \times \mathbf{T_R}$$

$$\mathbf{T_M} = \begin{bmatrix} \frac{L}{M} & j\omega\left(\frac{L^2}{M} - M\right) \\ \frac{1}{j\omega M} & \frac{L}{M} \end{bmatrix}, \mathbf{T_R} = \begin{bmatrix} 1 & R \\ 0 & 1 \end{bmatrix}, \mathbf{T_C} = \begin{bmatrix} 1 & 0 \\ Z_S^{-1} + Z_B^{-1} + j\omega C & 1 \end{bmatrix}, \mathbf{T_W} = \begin{bmatrix} 1 & R_W \\ 0 & 1 \end{bmatrix},$$

where $Z_S$ and $Z_B$ are identical to the previous case.

## Optimizing the MI metamaterials' bandwidth

This is majorly determined by the mutual coupling between neighbor coils, which, in fact, is not reflected in $S_{11}$ measurements from a single coil, and instead can be practically indicated by $S_{21}$ measurements across an array (to include the coupling information). Here, a horizontal overlap of about 20 mm (which translates to about 15% of the length of our reference coil design) reaches the desired bandwidth and transmission profile (Supplementary Fig. 6).

## Silver ink materials and synthesis

"Elmer's Glue-All" Multi-Purpose Liquid Glue was used as a water-based polymer matrix of polyvinyl acetate (PVAc). Silver flakes (SYP-981) with a mean particle size of around 10 μm were purchased from AG PRO Technology, Taiwan. Glyceryl triacetate (GTA, 99%), sodium

tetraborate decahydrate (borax, 100%), and acetic acid were purchased from Alfa Aesar Chemicals, Honeywell Fluka Chemicals, and J.T. Baker, respectively. SYLGARD®184 based polydimethylsiloxane (PDMS) was purchased from Merck KGaA. All chemicals, solvents, and reagents were analytical grade. Deionized water (DI water) was used in synthesis experiments.

## Synthesis of silver inks

For the preparation of pre-silver ink, PVAc (Glue-All = 41.3%), GTA, and acetic acid were first mixed by a mixer (THINKY MIXER AR-100) at 1000 rpm for 30 s. The acetic acid here provides a weak acid environment to prevent the ink from curing. Next, the silver flake powder was added into the adhesive agent with silver nanowire solution, which increases the conductivity and elongation of the ink, and then mixed for 3 min. For the final silver ink, borax (5% in water, 1% of PVAc + GTA) was added into the pre-silver ink and mixed for 3 min (see Supplementary Fig. 14).

## Stretchable resonator fabrication

The designed patterns were laser-cut (Versa laser, 1060 nm) onto disposable adhesive vinyl masks adhered to the desired Polydimethylsiloxane (PDMS) membrane substrate. After the conventional coating process, applying the silver ink over the stencil, the mask was then removed to reveal the coil pattern. The resonator coil traces were then developed by peeling the mask. Then, the surface mount tuning

capacitor (805 packages, 5% tolerance, 50 V rating, Murata Electronics) was placed on the dedicated area to make contact with the uncured silver ink. Next, the coil batch was cured in an oven under 100 °C for 30 min. This resulted in a 9.5 cm × 2 cm × 0.1 cm stretchable resonator on a 0.5 mm thick PDMS substrate (see Fig. 4a). Finally, the stack-up was sealed by transparent medical grade 3 M Tegaderm film.

## Strain measurements

The electric conductivity of the silver ink-based strip (2 cm × 1 cm on a TPU substrate) was measured by a four-point probe meter (Loresta-GP MCP-T600, Mitsubishi Chemical). The strain was applied by a motorized linear actuator (Zaber X-NA08A25-E09), and then strained at $20\,\mu m\,s^{-1}$ until pasting their electrical strain-to-failure point (breaking point of up to 900% strain). During the straining process, resistance changes were continuously recorded using a precision LCR meter (Keysight Technologies E4980AL). A similar setup was used to measure $S_{11}$ resonance properties while recording the reflection coefficient seen from an aligned reader antenna close to the coil (see Fig. 2b, e, f).

## NFC sensor design

The passive sensors are based on NFC transponders (Texas Instruments "TI" RF430FRL153H) and were designed and fabricated on a flexible printed circuit board (Supplementary Fig. 8). These boards incorporate an integrated loop antenna in addition to off-the-shelf external resistive strain gauges. The chips were then programmed over the air to announce the ADC output (known as the sensor value) upon the reader's interrogation command under ISO15693. The programming was performed using TI's GUI to interface between the TRF7970A (mounted on MSP430G2553), and the transponder chip. We used an off-the-shelf bend sensor (Short Flex Sensor, Adafruit Industries LLC) embedded in our sensor boards.

## NFC reader design

The active reader is based on the NFC controller (NXP PN7161) and is designed and fabricated in a miniature form factor to be mounted on the Raspberry Pi Zero. Additionally, we incorporated a custom-designed onboard NFC loop antenna (with custom onboard matching network) on the NFC reader to align and couple with the epidermal resonators for maximal end-to-end transmission efficiency.

## NFC sensor encapsulation

A thin layer (<1 mm) of 1:10 PDMS was poured into a 6-in. petri dish and allowed to partially cure for approximately 20 min at 80 °C to create a sticky base layer for the flexible printed circuit board substrate. The flexible NFC sensor was positioned centrally on top of the sticky PDMS and covered in 2–3 mm of 1:10 PDMS before being allowed to fully cure at 80 °C.

## Bluetooth RSSI measurement

We utilized two identical custom beacons (based on Nordic nrf52840 Bluetooth 5.3 transceiver) with different advertising names whose RSSI was recorded regularly while connected to a hub (smartphone). The hub and one of the beacons (as a control experiment) were fixed and immobilized in dry condition, while the other beacon (as the test experiment) was submerged in water after 100 s. Unlike the NFC protocol, the Bluetooth RSSI is reported only when a beacon is in reach, therefore the RSSI is not collectible when communication fails underwater.

## Statistics and reproducibility

No statistical method was used to predetermine the sample size. No data were excluded, and the experiments were not randomized. The investigators were not blinded to allocation during experiments and outcome assessment.

## Reporting summary

Further information on research design is available in the Nature Portfolio Reporting Summary linked to this article.

## Data availability

All data supporting the findings of this study are available within the article and provided in the Source Data file. Any additional requests (including data sources for supplementary information) can be directed to the corresponding author. Source data are provided in this paper.

## Code availability

The code supporting the NFC readout within this paper is based on the NXP application note (AN12991) publicly available from NXP.

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

## Acknowledgements

This work was partially supported by the National Science Foundation through grant CBET-1928326, as well as the CAREER award through ECCS-1942364 received by P.T.

## Author contributions

A.H. conducted the analytical studies, electronic hardware design, and simulations; P.R. developed and synthesized the materials; A.H.A.Z. developed the hardware-software interface; A.H., P.R., A.H.A.Z. and A.E. conducted the experiments; A.H. and P.T. conceptualized the study; P.T., M.K. and F.K. commented on the paper; A.H. and P.R. wrote the paper; A.H. and P.R. contributed equally.

## Competing interests

M.K. has an equity interest in Vena Vitals and Makani Science, companies that may benefit from the research results. The terms of this arrangement have been reviewed and approved by the University of California, Irvine in accordance with its conflict-of-interest policies. The remaining authors declare no competing interests.
