## [Peer Review File · Nature Communications]

REVIEWER COMMENTS

Reviewer #1 (Remarks to the Author):

--- separates review prompts from responses

*** separates responses from other review prompts

Key Results—What are the key messages of the study?

This paper presents a wireless sensing system for use in wearable applications requiring resilience to moisture and stretching. The key elements of the system are inductive coils that are fabricated using silver ink encapsulated with thin PDMS sheets. The coils can be arranged on the skin of a person with physical overlap to create a near-field transmission chain for establishing a wireless body-area network for exchanging power and data in the 13.56 MHz industrial, scientific, and medical (ISM) frequency band. The authors present analysis of the systems mechanical and electrical performance under strain, bending, and immersion in lossy media. They then demonstrate an extreme use-case for the system by measuring arm stroke motions with a strain gauge from a swimmer in a pool.

Validity of the Data, Methodology, and Conclusions—How robust are the interpretations of the data and conclusions?

The data in the paper is clearly presented and the individual experiments and measurements overall appear valid with some important exceptions.

For example, the authors state the usefulness of this solution across environments, including dry outdoors. However, there is no data supporting how the system performs when a user sweats. Additionally, it is not clear for how long the system can remain submerged—the longest submersion presented seems to be from the pool application that is several minutes long. While this is not necessarily a problem on its own, one justification of this system provided by the authors in the introduction is its robustness compared to wired systems, which was not sufficiently demonstrated.

The introduction states the disadvantage of wireless technologies requiring batteries, and yet the reader for the reported system is highly dependent on having a battery as well. A more nuanced description of these architectural differences would help the reader to better understand the benefits of the proposed solution.

The swimming test is impressive, yet the measurements do not seem to fully support the robustness of the system. In Fig. 4(d) we can see that there are data drops from the packets (again, not necessarily problematic, but such drops would limit the useful applications). In Fig. 4(e) measurements of a Bluetooth system being immersed in water are shown; while I understand it is nice to include the figure for comparison of two technologies, it is not a very insightful comparison, since it is common knowledge that far-field RF propagation is extremely unfavorable underwater.

The specific absorption rate (SAR) simulations are lacking information on the setup and calculations, so it is difficult to assess their accuracy. It appears that a 450 micro-Watt port was placed on a chain of the MI metamaterial coils that are on a slab of muscle. The low simulated SAR values of 1 mW/kg appear small, though the authors are correct that significant margin still exists for their system.

Significance—How significant are the contributions for the field and related fields? If other papers compromise the manuscript’s significance, please provide relevant references.

The work in this paper is impressive, but the contributions do not seem particularly significant. Several papers already exist that demonstrate the use of magneto-inductive metamaterial coils for transferring power and data, including from one of the authors’ research groups demonstrating many of the same techniques, measurements, and conclusions for a textile-based system. This work expands the application of this work to flexible silver-ink based electronics, but such stretchable electronics as well are not novel. These previous works are already cited by the authors.

Analytical approach—How strong is the analytical approach?

The analysis of the MI metamaterial coils is interesting with respect to how the waves propagate across the chain of coils. Little explanation or further analysis is provided though, except saying that the magnetic coupling of the MI metamaterials offers lower sensitivity to lossy surroundings, which is also a well-known result.

Suggested Improvements—Additional experimental or data useful?

I really like the idea for the system that the authors have created, and the coil propagation method is particularly interesting. The math leading to the dispersion equation seemed to be provided simply for the sake of having math, rather than providing more insight into the theoretical performance of the coils and the achieved performance when they are implemented using the silver ink and PDMS. Adding more information about this could improve the significance of the work.

The introduction is lacking references to other work on body-area networks. What would be the advantages/disadvantages of magnetic coupling through the body for a truly wireless solution? (e.g., DOI: 10.1109/TBME.2021.3101766)

Clarity and Context—How clear is the text? Is there sufficient context and consideration of previous work?

Overall the text is clear and the authors did a very good job designing clear, aesthetically pleasing figures.

Several typos, grammatical errors, and inconsistencies exist in the text.

References—Does the manuscript reference previous literature appropriately?

In the introduction, the authors seem to imply that a wired system from ref. 13 was limited to controlled clinical settings due to the difficulty of network expansion and the lack of mobility granted to users. I was unable to find the reference, and additionally, a system designed for controlled clinical applications is likely designed for a very specific use-case that may not call for network expansion or mobility.

The references in the introduction lack discussions about other types of body-area networks, seemingly focusing only on communication systems.

Relevant references were included in the sections above.

Reviewer #2 (Remarks to the Author):

The authors have presented an interesting work on an epidermal implementation of magneto inductive metamaterials in the form of stretchable skin patches for potential body area network application in communication and power transfer application. The proposed implementation has been proven to operate in different environments, and this has been demonstrated via the amphibious application of the proposed system. I believe that the experimental methods have been performed in a plausible way, resulting in data which are properly validated. Nonetheless, I believe there are several aspects that needs can be considered by the authors to improve their presentation, as follows:

- The main idea behind such MI implementation for BAN is not very different from the research team's previous work in ref. 31. There are of course several differences in terms of implementation, and the authors could explicitly list down these differences to help readers understand the significance of their work compared to their and other previous work.
- Another perhaps unclear statement by the authors is this: "Here, we demonstrate epidermal MI metamaterial networks to realize a body area network that is directly painted on the skin and functions seamlessly and regardless of the user's choice of clothing." However, when reading further, the implemented MI is actually not printed directly on the skin, if this is understood correctly, but instead on a thin non-conductive insulator. This reviewer understands that this can be improved if the lossy behavior of the loop traces can be controlled, but the claim can also be more accurately defined.
- Another question on the same statement, about the seamless function of the MI MTM regardless of the user's choice of clothing. This reviewer wonders about the effects of metallic accessories or if there would be any clothing with conductive-thread based material possibly worn in the proximity or even

over the proposed MI structure. How seriously will the coupling to the magnetic fields will this anticipated to be?

- Could the authors comment on how the thickness of the non-conductive insulators (top and bottom) will help in reducing the strain failure, and the tradeoffs with the performance indicated in Fig 2b and Supplementary Fig 1?
- It may be a good idea to add in the chosen insulator thickness into Fig 2c, as there are several thicknesses evaluated from Fig 2b. This is so that the readers don't need to search for the chosen 0.5 mm thickness (if this is indeed true) at the end of the manuscript (method section).
- A thickness value that seems unclear is the thickness/distance of the overlap between two coils, which is critical in enabling the adjustment of the pass bandwidth. Is this also designed to be the same as the thickness of the bottom substrate?
- What is the thickness of the 3M Tegaderm film mentioned in the "Methods: Stretchable resonator fabrication" section? And is this only used as the top-most sealant, or is it currently being used as the thin separator between the two overlapping coils?
- When designing the overlapping coils for the bandwidth of 848 kHz, could the authors guide this reviewer (and eventually the readers) on how this pass band is determined from the S11 in Fig 2c or other equivalent S11 figures?

Reviewer #3 (Remarks to the Author):

The authors present the development and assessment of a stretchable NFC sensors network that is capable of operating in dry and wet environments. The topic is very relevant to the community and the paper is well written, clear and technically sound. Yet, the novelty seems rather scarce. In fact, there is little innovation in the NFC sensors, although the authors do emphasize its robustness to the conductivity of the surrounding environment. Moreover, very few details on the simulation and design of the sensors are given. Given the aim of the Nature Communications for significant advancement papers, the reviewer does not recommend the publication of the manuscript as is.

Amphibious Epidermal Area Networks for Uninterrupted Wireless Data and Power Transfer:

Response to Reviewer Comments

Brief list of changes:

We genuinely value the constructive feedback provided by the reviewers and extend our gratitude to both the reviewers and the editorial team for dedicating their time and considering our manuscript. The insights and recommendations offered by the reviewers have been instrumental in enhancing the quality of our manuscript, and we believe that the revisions made in response to their input effectively address their concerns.

This document includes our detailed point-by-point response to comments, in addition to a brief list of changes applied in the revised manuscript in order of appearance:

- More detailed and systematic mathematical support for nearfield MI propagation
- Performed several new experimental studies including:
 - Insulator thickness optimization
 - Long-term epidermal network functionality
 - Reliability performance of the BAN
 - Strain tolerance versus encapsulation settings
 - Impact of sweat and metallic objects on network stability
- Implemented major enhancements in a few main figures
- Updated FEM simulations with additional details on SAR calculations
- Cited additional references on BAN and conductive ink synthesis
- Implemented multiple minor improvements for better readability and accuracy
- Updated Methods section with new details on network architecture, electronic hardware design and mathematical approaches
- Added several additional supplementary figures to represent new studies and hardware architectural novelties
- Presented supplementary tables of comparison to highlight the practical differences between this study and available solutions

Reviewer #1:

This paper presents a wireless sensing system for use in wearable applications requiring resilience to moisture and stretching. The key elements of the system are inductive coils that are fabricated using silver ink encapsulated with thin PDMS sheets. The coils can be arranged on the skin of a person with physical overlap to create a near-field transmission chain for establishing a wireless body-area network for exchanging power and data in the 13.56 MHz industrial, scientific, and medical (ISM) frequency band. The authors present analysis of the systems mechanical and electrical performance under strain, bending, and immersion in lossy media. They then demonstrate an extreme use-case for the system by measuring arm stroke motions with a strain gauge from a swimmer in a pool.

The data in the paper is clearly presented and the individual experiments and measurements overall appear valid with some important exceptions.

For example, the authors state the usefulness of this solution across environments, including dry outdoors. However, there is no data supporting how the system performs when a user sweats.

Response- We appreciate the reviewer's thorough evaluation of our manuscript.

Originally, to simulate sweating conditions in our study, we conducted experiments where saline (35 g/L NaCl solution) was gradually sprayed on the skin patches (until fully submerged underwater), mimicking the presence of moisture due to sweating. The results of these experiments were presented in **Fig. 2b** (comparing the path loss under dry and saline background with appropriate encapsulation) as well as **Fig. 4d** (which shows the digital end-to-end performance of the wireless sensing system under saline exposure).

Additionally, we also characterized the transmission profile of the MI metamaterial array when covered in saline, to assess its behavior in comparison to dry conditions. This scenario mimics sweating where the skin patches are exposed to saline. This analysis was conducted under various mechanical scenarios to evaluate the system's performance in realistic conditions when the user is in motion or subject to strain. The corresponding results were presented in **Fig. 3b**, illustrating the system's robustness in transmitting power and data even when exposed to saline, emulating sweating scenarios. We would like to highlight that the epidermal skin patches were securely fixed on the skin via off-the-shelf medical film (3M tegaderm), which are known to absorb saline up to 8 times their weight¹.

However, to further investigate the real sweating scenarios and address the reviewer's feedback in accurate details, we implemented an experiment to monitor the network PRR during an outdoor activity. We induced sweating through 5 min of outdoor pushups under 86°F, which is shown to impact the packet loss. The PRR slightly drops from 97% to 93% (from start to end of intense activity) which is primarily due to sweat droplets accumulated under the coils. This is comparable to underwater setting in which the network is surrounded in lossy media. Either way, the impact is relatively slight (with a maximum measured packet loss of 7%) and is expected to last until the network is completely dried. Supporting results are presented in the new **Supplementary Fig. 12a**.

Additionally, it is not clear for how long the system can remain submerged—the longest submersion presented seems to be from the pool application that is several minutes long. While this is not necessarily

¹ <https://multimedia.3m.com/mws/media/1851431O/3m-tegaderm-chg-dressings-faq.pdf>

a problem on its own, one justification of this system provided by the authors in the introduction is its robustness compared to wired systems, which was not sufficiently demonstrated.

Response- In response to this comment, we have taken steps to further demonstrate the system's durability and robustness under prolonged submersion. In the revised manuscript, we included an additional long-term monitoring test that involves capturing data from an underwater network for about 110 minutes, providing a substantial extension of the submersion duration compared to the pool application scenario. Supporting results and discussions are presented in **Supplementary Fig. 11**.

We believe that this addition will effectively showcase our system's ability to maintain reliable functionality over an extended period of time while submerged. This demonstration aligns with the key justification presented in the introduction regarding the system's robustness in comparison to wired systems. Our intention is to provide a comprehensive perspective on the system's performance and durability, further substantiating its practical applicability and resilience in challenging environments.

The introduction states the disadvantage of wireless technologies requiring batteries, and yet the reader for the reported system is highly dependent on having a battery as well. A more nuanced description of these architectural differences would help the reader to better understand the benefits of the proposed solution.

Response- We appreciate the reviewer's perspective and would like to provide a more detailed description of the architectural differences between conventional wireless technologies and our proposed solution.

While it is true that wearable devices, including our reported system, require a power source such as a battery or power harvesting mechanism, it is important to emphasize that the innovation roots in how power is distributed and utilized. The key distinction lies in establishing a reliable power distribution channel from the source to various passive electronic loads distributed across the body. In other words, standalone power harvesting mechanisms that do not rely on batteries, can also yet benefit from this NFC compatible BAN strategy that can efficiently distribute harvested power to distributed loads, especially in challenging-to-reach places where lightweight solutions are crucial.

Our epidermal skin patches introduce a passive pathway for power distribution. This distinctive feature allows multiple smaller electronic loads to operate without the need for individual batteries. This design choice offers significant advantages in terms of integration into the dynamic environment of the human body. By eliminating the requirement for batteries in these smaller passive loads, the wearables become more adaptable to the body's movement and fluctuations.

Furthermore, this architecture liberates the wearables from the limitations of frequent charging cycles that necessitate repeated plug/unplug actions. The power distribution network created by the skin patches ensures a continuous flow of power to the loads, alleviating the need for users to frequently interact with the devices for charging purposes.

Additionally, we added the architectural hardware comparison between the traditional wired and the proposed wireless platform in **Supplementary Fig. 13**, along with a detailed comparison between available BAN technologies in **Supplementary Table 1**. Supporting discussions are added to the closing paragraphs of **section 4** revised manuscript.

The swimming test is impressive, yet the measurements do not seem to fully support the robustness of

the system. In Fig. 4(d) we can see that there are data drops from the packets (again, not necessarily problematic, but such drops would limit the useful applications). In Fig. 4(e) measurements of a Bluetooth system being immersed in water are shown; while I understand it is nice to include the figure for comparison of two technologies, it is not a very insightful comparison, since it is common knowledge that far-field RF propagation is extremely unfavorable underwater.

Response- We acknowledge the presence of minor data drops from the packets (shown in **Fig. 4d**), which may raise concerns about the system's robustness. However, it's important to note that the packet loss in our system is exceptionally low, measuring less than 5% with regards to this experiment conditions. This level of packet loss is well within acceptable limits for many practical applications, particularly human monitoring purposes with relatively slow rate of action. Moreover, we have implemented measures to compensate for any lost packets by slightly increasing the sampling rate. This approach ensures that even with occasional packet drops, our system's data collection reaches the Nyquist sampling rate for a comprehensive range of human activities. Additionally, we would like to highlight the new studies (in **Supplementary Fig. 11**) to demonstrate the robustness of the network in underwater conditions.

Originally, we included the measurements of a Bluetooth system immersed in water (**Fig. 4e**) for a fair comparison with our BAN, recognizing that far-field RF propagation underwater is challenging for such technologies. While it may seem like a common knowledge comparison, it serves to emphasize a key point that despite the challenges associated with underwater RF propagation, the MI BANs remains fully functional. The notable distinction here is that our system achieves successful communication and data transfer underwater, whereas conventional Bluetooth technology experiences 100% packet loss in such scenarios. This is in fact an entirely new horizon for wireless power and data transmission, and after careful consideration, we believe **Fig. 4e** aligns with our initial concept demonstration in **Fig. 1c**, and decided to keep it in this main figure of the revised manuscript.

The specific absorption rate (SAR) simulations are lacking information on the setup and calculations, so it is difficult to assess their accuracy. It appears that a 450 micro-Watt port was placed on a chain of the MI metamaterial coils that are on a slab of muscle. The low simulated SAR values of 1 mW/kg appear small, though the authors are correct that significant margin still exists for their system.

Response- In response, the reviewer is correct that the encapsulated MI metamaterial is placed on a muscle slab (with thickness that is significantly larger than the skin depth in muscle at 13.5 MHz). For a general evaluation, we originally adjusted the NFC reader's output power (450 μ W) to meet the maximum allowable emitted magnetic field permitted by ISO/IEC 10373-6 (3 A/m) for the multiturn loop antenna of this simulation.

In the revised manuscript, we adjusted our simulation in order to meet the worst-case scenario in which our NFC reader chip (NXP PN7161) reaches the maximum emitted power (this may not be necessarily NFC compliant). According to section 4 of <https://www.nxp.com/docs/en/application-note/AN13224.pdf>, the dynamic power control of this NFC reader chip adjusts the switched voltage and load current to 2.8 V and 200 mA, respectively, resulting in 560 mW of RF available power. We should however, note that this indicates the maximum allowable RF power, and practically this is never achieved due to the amplitude modulation in an NFC compliant active reader. This new worst-case simulation setting resulted in the maximum average SAR of 82 mW/kg, which is yet far from the common safety thresholds.

New **Supplementary Fig. 7** includes updated SAR simulations and addresses details regarding the material properties, port and insulator's position. In addition, the corresponding paragraph in the manuscript body now addresses this concern.

The work in this paper is impressive, but the contributions do not seem particularly significant. Several papers already exist that demonstrate the use of magneto-inductive metamaterial coils for transferring power and data, including from one of the authors' research groups demonstrating many of the same techniques, measurements, and conclusions for a textile-based system. This work expands the application of this work to flexible silver-ink based electronics, but such stretchable electronics as well are not novel. These previous works are already cited by the authors.

Response- We understand and acknowledge this point, and appreciate the reviewer for highlighting that. We would like to emphasize that we have introduced several layers of novelty to the domain of BAN, and demonstrated the evolution of existing and newly developed techniques to enable wearable electronics enter new horizons that have not been accessible before—this includes the preliminary studies performed either by our team or other research.

For instance, MI propagation has been introduced in early 2000s, however, its use has been strictly limited to wireless power transfer in highly controlled environments, and ranges that are a few folds smaller than our ecosystem, and particularly through micro-tuned rigid resonant structures. We earlier developed a new approach to harness this strong concept and enable it to be exploited for NFC in consumer electronics for the first time². The copper-based, textile-integrated BAN proposed in the aforementioned study, however, is fundamentally unable to be utilized directly on organic living tissues due to lack of stretchability that may not conform to complex body positions.

We believe this study introduces multiple dimensions of novelty from mathematical methods of modeling complex resonators (with serpentine geometry), to implementation of highly stretchable and excellently conductive resonators, and ultimately realizing extreme applications that have not been reachable ever since. In fact, not only the conventionally fragile MI metamaterial structures, but also RF communication systems have never tackled such challenges to the extent presented in this study.

To further address the reviewer's concern, we added **Supplementary Tables 1 and 2** to highlight the high-level fundamental differences between this study and existing state-of-art in terms of BAN technologies and conductive inks.

The analysis of the MI metamaterial coils is interesting with respect to how the waves propagate across the chain of coils. Little explanation or further analysis is provided though, except saying that the magnetic coupling of the MI metamaterials offers lower sensitivity to lossy surroundings, which is also a well-known result.

Response- We appreciate the reviewer's constructive observation regarding the dispersion equation and its role in enhancing the theoretical and practical understanding of the coils' performance.

In response to the reviewer's input, we have revised the manuscript to provide a more comprehensive analysis of the coils' behavior. Specifically, we have introduced a systematic approach based on a piecewise cascaded transfer matrix method. This approach not only validates the traditional MI

² Hajiaghajani, A, et al., "Textile-integrated metamaterials for near-field multibody area networks," Nat. Electron., 2021.

propagation equation but also enables a more detailed exploration of complex coil structures, such as those incorporating serpentine patterns and encapsulated in PDMS sealing within lossy settings.

For this purpose, we have renewed our mathematical analysis throughout the manuscript (that verifies the traditional approach) and heavily modified **Fig. 1b** and **Supplementary Fig. 2**.

I really like the idea for the system that the authors have created, and the coil propagation method is particularly interesting. The math leading to the dispersion equation seemed to be provided simply for the sake of having math, rather than providing more insight into the theoretical performance of the coils and the achieved performance when they are implemented using the silver ink and PDMS. Adding more information about this could improve the significance of the work.

Response- We appreciate the reviewer's comment, and would like to highlight that in response to the last comment, we heavily updated the mathematical analysis by breaking the complex resonator (that may possess serpentine-shaped structure, encapsulated in PDMS, and placed on the skin while submerged in a lossy medium) into sub-models whose transfer matrices are cascaded. Unlike the traditional method of analyzing the MI metamaterials (through deriving the loop impedance and using Kirchhoff's circuit laws in the n^{th} resonator), this approach powerfully enables systematic extraction of dispersion diagrams for such waveguides. By calculating the transfer matrix of the resonator's unit cell (T_{cell}), one may calculate the end-to-end transfer matrix of a linear MI array of N resonators (T_{cell}^N), and simply derive the transmission profile in form of S-parameters³. This approach offers a significantly more robust analysis, as in contrast to traditional MI array modeling, we do not assume an unrealistically large number of resonators within the chain.

By incorporating this systematic approach, we aim to offer a more thorough and insightful investigation into the theoretical performance of the coils. This deeper analysis contributes not only to a better understanding of the underlying principles but also to a more robust framework for the practical implementation of the coils using silver ink and PDMS. We have accordingly added the detailed content in **Fig. 1b** and **Supplementary Fig. 2**.

The introduction is lacking references to other work on body-area networks. What would be the advantages/disadvantages of magnetic coupling through the body for a truly wireless solution? (e.g., DOI: 10.1109/TBME.2021.3101766)

Response- We sincerely appreciate the reviewer's comment, bringing this publication to our attention. As suggested by the reviewer, we believe the magnetically-dominant nearfield human body communication solution proposed in this reference is closely comparable to our study. Importantly, we would like to highlight that one key factor in versatility of BAN solutions is compatibility with pre-existing standards, in addition to compliance with off-the-shelf hardware and underlying software libraries. While the magnetically coupled nearfield antennas of this reference is significantly more robust compared to BLE, the available hardware modules and software interfaces may limit its usage in emerging wearable applications that aim to minimize end user costs while integrating multitude third-party sensors and devices. This point of view highlights the importance of NFC compliance in our study, as our skin patches are readily functional with any smartphone and any NFC transponder chips.

³ Pozar, David M. *Microwave Engineering*. 3rd ed, J. Wiley, 2005

To further address the reviewer's concern, we added **Supplementary Tables 1** to highlight the high-level fundamental differences between this study and existing state-of-art BAN technologies. We modified the introduction to reflect this reference and similar studies. Once again we appreciate this valuable feedback.

Overall the text is clear and the authors did a very good job designing clear, aesthetically pleasing figures. Several typos, grammatical errors, and inconsistencies exist in the text.

Response- We appreciate the reviewer's thorough comments. We revised several areas of the manuscript for better readability and improved consistency.

In the introduction, the authors seem to imply that a wired system from ref. 13 was limited to controlled clinical settings due to the difficulty of network expansion and the lack of mobility granted to users. I was unable to find the reference, and additionally, a system designed for controlled clinical applications is likely designed for a very specific use-case that may not call for network expansion or mobility.

Response- This reference points to the US patent associated with the first FDA-approved body area network (Radius VSM™) designed and developed by Masimo Corp., a world-class biomedical device company headquartered close to our UC Irvine campus. We revised this citation to ensure accessibility. More information can be found on <https://www.masimo.com/products/continuous/radius-vsm/>

The references in the introduction lack discussions about other types of body-area networks, seemingly focusing only on communication systems. Relevant references were included in the sections above.

Response- We would like to thank the reviewer again for noting these references. We added more than 10 new references focusing on different but comparable BAN technologies, ultimately showcased in **Supplementary Table 1** as well as the revised introduction.

Reviewer #2:

The authors have presented an interesting work on an epidermal implementation of magneto inductive metamaterials in the form of stretchable skin patches for potential body area network application in communication and power transfer application. The proposed implementation has been proven to operate in different environments, and this has been demonstrated via the amphibious application of the proposed system. I believe that the experimental methods have been performed in a plausible way, resulting in data which are properly validated. Nonetheless, I believe there are several aspects that needs can be considered by the authors to improve their presentation, as follows:

- The main idea behind such MI implementation for BAN is not very different from the research team's previous work in ref. 31. There are of course several differences in terms of implementation, and the authors could explicitly list down these differences to help readers understand the significance of their work compared to their and other previous work.

Response- We would like to thank the reviewer for evaluating our study and providing us with valuable comments that led us to enhancing the quality of our presentation. We agree with the reviewer that the core physical mechanism behind the textile-integrated NFC compatible MI waveguides is rooted in our earlier study, however, we would like to emphasize that our previous work inherently lacks the capability to function effectively in the extreme environments presented in this current research.

Specifically, our earlier approach was limited in its mechanical stretchability due to the use of copper-based materials. In contrast, the primary motivations behind this study were twofold: firstly, the imperative need for maximal mechanical stretchability, which we have now achieved, and secondly, the introduction of entirely new and challenging environments in which wearable electronics can operate. These novel environments, such as epidermal implementation and highly lossy media, set this work apart.

We believe this study introduces multiple dimensions of novelty from mathematical methods of modeling complex resonators (with serpentine geometry), to implementation of highly stretchable and excellently conductive resonators, and ultimately realizing extreme applications that have not been reachable ever since. In fact, not only the conventionally fragile MI metamaterial structures, but also RF communication systems have never tackled such challenges to the extent presented in this study. The significance of our research lies not only in enhancing the reliability of wearable electronics for critical applications but also in broadening the horizons for their utility in previously unexplored domains. This expansion extends the reach of wearable electronics, making them valuable in a continuous and uninterrupted manner.

We would like to point out that in many BAN-focused studies, the core technologies are not necessarily brand new innovations on their own. Rather, the ingenuity lies in the way these established concepts are harnessed to extend their applications beyond their original scope. For instance, several textile-integrated BANs rely on well-established principles from the field of applied electromagnetics, such as different types of metamaterials that are made compatible with existing protocols such as BLE⁴. However, the innovative synthesis of these well-known techniques and concepts in novel combinations that have not been previously explored sets these endeavors apart.

Finally, to further clarify the distinctions and emphasize the significance of this work, we have included a **Supplementary Table 1** in the revised manuscript. This table not only provides a comparative analysis of our research outcomes with other existing BAN solutions but also highlights the unique contributions and advantages over our previous study in terms of performance and adaptability in extreme

⁴ Tian, X. et al. "Wireless body sensor networks based on metamaterial textiles". Nat. Electron. (2019)

environments. We believe that this addition will enhance the understanding of the distinctive features and significance of our work compared to previous research efforts.

• Another perhaps unclear statement by the authors is this: “Here, we demonstrate epidermal MI metamaterial networks to realize a body area network that is directly painted on the skin and functions seamlessly and regardless of the user’s choice of clothing.” However, when reading further, the implemented MI is actually not printed directly on the skin, if this is understood correctly, but instead on a thin non-conductive insulator. This reviewer understands that this can be improved if the lossy behavior of the loop traces can be controlled, but the claim can also be more accurately defined.

Response- The reviewer is correct in that the coils are actually painted on the insulator layer (and then directly “placed” on skin rather than “painted”) and we appreciate them for highlighting it. We originally intended to emphasize that the skin patches are not attached to clothing or nearby peripherals, and are readily usable on bare skin. Additionally, printing the resonators on thin insulator layers (forming skin patches) offers better versatility, precise coil geometry control that is ideal for wearable usage, providing with a natural interface similar to wound patches. Moreover, this strategy offers prolonged utilization of skin patches due to reusable encapsulation. This is edited in the revised manuscript for improved statement accuracy.

• Another question on the same statement, about the seamless function of the MI MTM regardless of the user’s choice of clothing. This reviewer wonders about the effects of metallic accessories or if there would be any clothing with conductive-thread based material possibly worn in the proximity or even over the proposed MI structure. How seriously will the coupling to the magnetic fields will this anticipated to be?

Response- We genuinely appreciate this comment, as we believe it touches one of the main advantages of this mechanism over other BAN solutions. We would like to start elaborating this from an electromagnetic point of view in our waveguide’s unit cell equivalent circuit (**Fig. 1b** and **Supplementary Fig. 2**).

Adding a metallic object or a perfect electric conductor (PEC) as in the worst case scenario would bypass the fringe electric fields by offering a low loss shortcut path, which can be modeled by shorting the fringe capacitors (C_B). This may not only potentially shift the resonance characteristics of the coil, but also may diminish the transmission profile of the MI array. This, in fact, justifies the need for encapsulating the resonators with appropriate insulation properties which is modeled as R_{INS} . However, due to mechanical limitations on insulator thickness, the insulation may not be ideal, hence a finite R_{INS} value would be inevitable. This induces some level of resonance sensitivity to external metallic objects, but one may expect only modest perturbations due to series R_{INS} . This further on justifies use of lumped capacitor (shown as C), to ensure its admittance dominates that of the background medium, therefore ensuring a stable resonance characteristics even under extreme settings such as nearby PEC or significantly lossy media. Larger R_{INS} (i.e. thicker encapsulation) and lumped C values warrant better immunity to such interferences. This translates to allowing fringing electric fields form within the lossless insulator’s thickness.

Additionally, to evaluate the end-to-end performance of the BAN under such setting we performed a new study (demonstrated in new **Supplementary Fig. 12b**), and as expected, observed a very slight decline in PRR with a loss rate of 7 packets/min when a metallic object (here an aluminum sheet) covers about 30 cm² of the dry encapsulated skin patches. We summarized and added this discussion to **Section 2** of the revised manuscript.

- Could the authors comment on how the thickness of the non-conductive insulators (top and bottom) will help in reducing the strain failure, and the tradeoffs with the performance indicated in Fig 2b and Supplementary Fig 1?

Response- The insulator thickness was chosen based on the measured transmission profile, to ensure minimum insertion loss (per unit of length) is achieved. This includes comparison among arrays of resonators encapsulated by 0.25, 0.5, and 0.75 mm thick PDMS sheets in dry and submerged in saline solution (35 g/L mimicking ocean's salinity). Supporting results are presented in **Fig. 2b** in the revised manuscript.

According to this comparison, the MI channel's insertion loss at 85 cm distance (between the Tx and Rx ports) is measured 37, 39, 40, and 44 dB for dry (0.25mm thick), saline (0.5mm thick), saline (0.75mm thick), and saline (0.25mm thick), respectively. According to this study, in saline setting, a significant improvement of 4 dB is observed between 0.25 and 0.5 mm PDMS thickness, however this reduces to 1 dB between 0.5 and 0.75 mm thickness. To maintain a balance between the resonator's thickness (that directly impacts flexibility thus user comfort) and channel's transmission performance, we chose 0.5 mm thick PDMS sheets for encapsulation.

From a mathematical perspective, the insulator thickness directly affects $R_{INS} = \frac{1}{\sigma_{INS}} \times \frac{\text{insulator thickness}}{\text{effective coverage area}}$ whose effect has been studied in **Supplementary Fig. 1b**. We accordingly added these discussions to the manuscript, and would like to thank the reviewer for bringing this point to our attention.

- It may be a good idea to add in the chosen insulator thickness into Fig 2c, as there are several thicknesses evaluated from Fig 2b. This is so that the readers don't need to search for the chosen 0.5 mm thickness (if this is indeed true) at the end of the manuscript (method section).

Response- We would like to thank the reviewer for helping us improve the readability of the manuscript. Yes, we chose 0.5 mm thick PDMS insulator, and added this information in caption of **Fig. 2c** to ensure clarity.

- A thickness value that seems unclear is the thickness/distance of the overlap between two coils, which is critical in enabling the adjustment of the pass bandwidth. Is this also designed to be the same as the thickness of the bottom substrate?

Response- The vertical distance between neighbor coils is determined by the thickness of cover (of underneath resonator) plus substrate thickness of the overlapping coil (on top). Although the mutual coupling between neighbor resonators directly impacts the bandwidth, when encapsulation thickness is small enough (less than about 1 mm here) the coupling is majorly dependent on the horizontal overlap (i.e. coil distancing) rather than the vertical distancing. In addition, the newly discussed results (supported in **Fig. 2b**) also confirms that the sealing thickness (represented as R_{INS} in **Fig. 1b**) does not play a noticeable effect on the bandwidth as well, as long as it is not dramatically thinned (this is supported in **Supplementary Fig. 1**).

- What is the thickness of the 3M Tegaderm film mentioned in the "Methods: Stretchable resonator

fabrication” section? And is this only used as the top-most sealant, or is it currently being used as the thin separator between the two overlapping coils?

Response- Our measurement shows a Tegaderm thickness of less than 80 μm , which is thinner than the designed patch cover (which as described in above comments, is added to create sufficient isolation for resonator’s fringing electric fields). We secured each resonator by a separate pieces of tegaderm such that the entire array is not covered by a single coat and therefore is easier to route. As a result, the neighbor coils are vertically separated by a twice the thickness of the PDMS substrate to ensure each coil is electrically immune from the lossy background environment. Accordingly, the coils are separated by 1 mm (two 0.5 mm thick separate PDMS layers). These results are supported in the new **Supplementary Fig. 6** in the revised manuscript.

• When designing the overlapping coils for the bandwidth of 848 kHz, could the authors guide this reviewer (and eventually the readers) on how this pass band is determined from the S11 in Fig 2c or other equivalent S11 figures?

Response- The passband bandwidth in MI metamaterials are majorly determined by the mutual coupling between neighbor coils, which, in fact, is not reflected in S11 measurements from a single coil, and instead can be practically indicated by S21 measurements across an array (to include the coupling information). We found that a horizontal overlap of about 20 mm (which translates to about 15% of the length of our reference coil design) reaches the desired bandwidth. We originally determined this passband based on experimental measurements, which are in great agreement with our mathematical analysis (which includes electrical characteristics of the multiturn loop as well). Generally, this threshold depends on the custom resonator’s geometry (that impacts self inductance of L, and therefore mutual inductance of M), however a coupling factor of about 0.1 has sufficed in our case. Additionally, we added the transmission profile per resonator under various levels of stress in **Supplementary Fig. 5**.

Reviewer #3:

The authors present the development and assessment of a stretchable NFC sensors network that is capable of operating in dry and wet environments. The topic is very relevant to the community and the paper is well written, clear and technically sound. Yet, the novelty seems rather scarce. In fact, there is little innovation in the NFC sensors, although the authors do emphasize its robustness to the conductivity of the surrounding environment. Moreover, very few details on the simulation and design of the sensors are given. Given the aim of the Nature Communications for significant advancement papers, the reviewer does not recommend the publication of the manuscript as is.

Response- We would like to thank the reviewer for evaluating our manuscript and providing us with valuable feedback.

It is in fact accurate to note that the novelty within NFC sensor and actuator design itself may appear limited, given that our custom electronics are based on established NFC chip manufacturer guidelines. However, it is essential to recognize the broader context of sensor development, which lies behind the data/power transmitting channels from/to these devices. Over recent years, the sensor field has seen remarkable progress, resulting in highly refined analog and digital sensors that leave little room for substantial improvements in their core functionalities. In this context, the focus of the wearable electronic community has shifted towards addressing critical challenges, such as minimizing power consumption, optimizing user comfort, and establishing seamless data flow between organic media or living bodies and computer cloud systems.

Our study contributes significantly to this ongoing paradigm shift. We extend the capabilities of one of the most prevalent communication protocols, NFC, beyond its traditional scope. Notably, we enable the creation of exceptionally lightweight and miniature sensor boards that operate without the need for batteries, are relatively distanced apart, and continue to function in extreme settings, because of the proposed epidermal BAN mechanism.

Comparative analysis, (presented in **Supplementary Table 1**), highlights the distinctive innovations introduced by our approach, particularly in the realm of power harvesting BANs. Our work bridges the gap between the sensor community and the potential of protocols like NFC in applications that were previously unattainable due to their limited range. This expansion opens new horizons for wearables, allowing them to thrive in exceptionally challenging environments, such as underwater settings. The uniqueness and comprehensiveness of our study in this regard are evident, and to our knowledge, there are no comparable reports of wireless power and data link in extreme settings to the extent demonstrated in this study.

We have enabled flexibility to integrate a wide range of sensors and actuators into this versatile amphibious channel framework that empowers pre-existing hardware to function seamlessly in entirely novel and challenging environments. This adaptability enhances the potential for innovation and application across various domains, and is in fact the focus of our study.

Moreover, we designed the flexible NFC sensor board (utilizing Texas Instruments RF430FRL153H) and miniaturized NFC reader (utilizing NXP PN7161) (with layouts available in **Supplementary Fig. 8**) based on the chip manufacturer recommended design, available in **Methods**. We incorporated an onboard NFC loop antenna (with custom onboard matching network) on the NFC reader that mounts on a Raspberry Pi Zero. We added extra details on NFC sensors and reader in **Methods**. The schematic designs of these boards are available from the chip manufacturer evaluation boards, and are available from the corresponding author upon request.

REVIEWERS' COMMENTS

Reviewer #1 (Remarks to the Author):

I would like to thank the authors for their thorough review of the paper. They have invested a lot of time and energy to sufficiently address all of my review comments.

Reviewer #2 (Remarks to the Author):

The authors have addressed most of my comments in a satisfactory manner, and I appreciate this.

Reviewer #3 (Remarks to the Author):

The authors have addressed the novelty concern pointed out by the reviewer, who does not have any additional comments.